# The Role of Emotional Intelligence and Metacognition in Teachers' Stress during Pandemic Remote Working: A Moderated Mediation Model

Calogero Iacolino [1], Brenda Cervellione [1], Rachele Isgrò [1], Ester Maria Concetta Lombardo [1], Giuseppina Ferracane [1], Massimiliano Barattucci [2,*] and Tiziana Ramaci [1]

1 Faculty of Human and Social Sciences, Kore University of Enna, 94100 Enna, Italy
2 Department of Human and Social Sciences, University of Bergamo, 24129 Bergamo, Italy
* Correspondence: massimiliano.barattucci@unibg.it

**Abstract:** During the COVID-19 pandemic, in adapting to social and work changes and new technological methods for remote teaching, teachers were subjected to increased work pressure, which affected their well-being and led to increased negative stress and burnout. This study was designed to test whether dysfunctional outcomes resulting from adapting to new ways of teaching via technological tools can be mitigated by the protective factors of emotional intelligence and metacognition. The study involved 604 teachers in Sicily filling out a questionnaire consisting of four different scales: (1) the Schutte Self-Report Emotional Intelligence Test (SREIT); (2) the Metacognitive Functions Screening Scale (MFSS-30); (3) the Link Burnout Questionnaire (LBQ); and (4) the Secondary Traumatic Stress Scale STSS-I. The results show that emotional intelligence mediates the relationship between certain remote work risk factors, as well as stress and burnout. In addition, metacognition was found to be a significant moderating factor in the relationship between risk factors and emotional intelligence. With regard to the United Nations' Agenda 2030 Goals, our results emphasize the importance of teachers' emotional and metacognitive skills in promoting quality of life and psychological well-being.

**Keywords:** teachers; remote working; COVID-19; emotional intelligence; metacognition; burnout; stress

## 1. Introduction

The COVID-19 pandemic has generated psychological distress in a variety of working environments, including education. The literature [1,2] has shown that teachers accumulated various symptoms of negative stress and burnout [3] attributable to the introduction of governmental lockdown measures in Italy. Great concern emerged regarding the effects of the unprecedented online teaching situation [4], with remote work seen as a determining factor in the increase of psychological and clinical symptoms [5–7]. Indeed, being forced to quickly adapt to new ways of working, including hybrid and remote teaching [8–10], has been associated by teachers with increased levels of stress [11]. Such new stressors, which may lead to burnout, have only added to the fear generated by COVID-19, affecting almost all people in society [12,13].

According to the current psycho–physiological definition, stress is a notable, persistent condition in which an organism is exposed to risk factors that tend to alter its self-regulating balance, or homeostasis, which thus affects both individual and group healthy behaviors [14]. The consequences of stress and psychophysical exhaustion are various, and, in the teaching profession, may include significant professional setbacks, work absences, and reduced efficiency in class control and class work [15–17] and may even lead to abandoning the profession entirely [18–20]. Thus, stress represents a significant risk factor in the job satisfaction of teachers [21]. However, certain protective factors,

such as emotional intelligence and metacognition, may help mitigate the effects of stress on teachers.

According to the theoretical model of Salovey and Mayer [22], emotional intelligence (EI) in teachers is: (1) the ability to be self-aware of one's emotions and to be able to identify them; (2) the ability to distinguish among various types of emotions; (3) the ability to comprehend and analyze one's emotions; and (4) the ability to regulate emotions in oneself and in others. Such abilities are thought to constitute a protective factor in the management of educational activities, occupational stress, and personal exhaustion [23–27]. It has been shown [28] that teachers with high levels of emotional intelligence are more able to adapt to different personal life contexts, whether social or professional. On the other hand, low levels of emotional intelligence are correlated with psychopathological consequences such as stress, distress, and burnout [29]. It is therefore fundamental to complete everything possible to help teachers learn to regulate their emotions, particularly at the teacher training stage [28], in order to boost their cognitive, emotional, and social skills.

Another important protective factor in the work of teachers is metacognition (MC). This is defined as an aspect of the elaboration of information that controls, interprets, evaluates, and regulates content, all cognitive processes, and their organization [30]. It is an awareness that allows one to control one's own cognitive processes, favoring the ability to learn, remember, identify, and solve problems [31]. This ability allows one to give meaning to one's experience through a mental procedure and to understand oneself and others in terms of mental states, such as feelings, convictions, intentions, and desires, and to reflect on one's own behaviors and those of others [30].

Since many empirical and theoretical contributions concur that such protective factors can have a positive effect on work performance, productivity, and relationships, it is important to further understand their practical relevance in terms of healthy behavioral outcomes and occupational health and well-being [32]. It is thus with the aim of investigating the effects of EI and MC as protective factors on remote working stressors in particular and on health outcomes that this correlational research involving teachers was designed.

## 2. Literature Background

From March 2020, Italian teachers had to face many changes to their work activities, potentially leading to an increase in the perception of work stressors and the intensity teachers attribute to such stressors, making them even more prone to risks of maladjustment. Klapproth et al. [33] found that, on average, German teachers experienced moderately high levels of stress due to a lack of adequate technological equipment and internet connectivity, an excessive workload, and student demotivation, as both internal and external obstacles made remote teaching work difficult. In Argentina, Vargas Rubilar and Oros [34] described, among pandemic remote teaching occupational risks, there were five main stressors: (1) the work environment and work overload; (2) relationships with students; (3) conflict and role ambiguity; (4) organizational aspects of the educational institution; and (5) the use of new technologies. Concerning the latter stressor, the Confederation of Educational Workers of the Argentinean Republic [35] found that a high percentage of teachers reported not having adequate technological resources, nor a comfortable, isolated place to work from at home. Another consequence of remote working in the educational sector is the increased time devoted to online teaching [36]. Finally, other significant difficulties experienced by teachers are a lack of support among colleagues, poor coordination, and difficulties in teamwork [34].

Based on the empirical literature, the main remote working risk factors for teachers identified for this study were (1) a lack of adequate IT tools of exclusive use for online teaching; (2) a lack of computer skills in the use of online educational platforms; (3) a lack of an adequate place of exclusive use for online teaching; (4) a lack of IT skills and time for adapting to online teaching methods; (5) difficulties in managing the class during online teaching; and (6) difficulties in terms of coordination and support among colleagues. In view of such risk factors concerning remote working, the aim of this research was, therefore, to study

two factors that play a protective role against burnout risk in teachers [3,13,25,27,30,37], namely emotional intelligence and metacognition.

*The Role of Emotional Intelligence and Metacognition in Relation to Teacher Stress and Burnout*

The stressful experience of teachers having to work remotely is a widespread phenomenon of the COVID-19 pandemic. For example, recent reports on teachers' emotions have reported a range of associated unsatisfactory social relationships with adults, including colleagues, headmasters, parents, and inspectors, arousing hostile emotions in teachers and constituting a source of stress in teaching [38]. Furthermore, teacher stress has been seen as a response to negative organizational factors and teaching aspects being perceived as a threat to physical and psychological well-being [39].

The experience of a stressful teaching work environment also has clinical implications for burnout syndrome. Burnout has been defined as "a psychological syndrome emerging as a prolonged response to chronic interpersonal stressors on the job" [40] (p. 103) and, therefore, as an occupational psychological syndrome [41–43], often affecting medical professionals, teachers, police officers, army soldiers, and many other professionals [44–53]. Certain studies have detected relationships between burnout and EI [54,55], as an important personal resource that can facilitate effective emotion regulation [56] and play a crucial role in teachers' occupational well-being models [57]. Indeed, the literature seems to confirm that EI is a protective factor, since low levels of EI tend to be correlated with burnout [58] and psychological distress [59]. Furthermore, some authors argue that teachers' perceptions of their own abilities to manage stress and influence effective emotional regulation [60]. Recently, studies have also shown that EI correlates with greater job satisfaction [61], and with greater satisfaction in teaching in particular [62]. According to some moderation models [54,63], teachers with high EI can better manage anger and frustration, while those with low EI levels experience increased tension in the student–teacher relationship [64]. Similarly, one study [59] has shown that teachers with low EI feel they are unable to cope with perceived stress in the workplace and, therefore, report professional ineffectiveness.

EI has been shown to have a statistically significant negative correlation with teacher burnout, with variables such as gender and age not appearing to have a moderating role in this correlation [65]. However, other studies [1] have identified sex differences in the experience of burnout, with female teachers in particular experiencing significantly greater emotional exhaustion and lower job fulfillment than men, who tend instead to experience depersonalization. Some research [29] shows that younger female teachers are more at risk of developing burnout than older teachers, while another study points out that younger teachers are more motivated and enthusiastic about their work than older teachers [28]. In any case, there is clear supportive evidence of a relationship between EI, perceived stress, and teacher burnout [66], with more effective emotional regulation acting as a moderating factor in managing perceived stress and preventing burnout.

MC, on the other hand, has been shown by a large body of literature to relate to symptoms of stress, anxiety, or depression [31], with one study, [30], showing that dysfunctional metacognition and metacognitive strategies negatively influence emotions and can predict traumatic stress symptoms. Dysfunctional beliefs and convictions about one's own mental processes can have an impact on behavioral and cognitive responses by influencing perceived stress and emotions [67]. Individuals who believe that their thoughts influence reality in both positive and negative ways are more likely to use metacognitive strategies for regulating emotions and thinking in relation to perceived stress. Indeed, teachers with good metacognitive and emotional skills have been observed to respond more productively during particularly stressful situations by showing greater resilience to stressors [68,69]. The construct of MC can be a particularly important factor in school contexts, as a greater perceived awareness of one's own metacognitive processes can produce significant effects in preventing clinical symptoms such as burnout and promote the cognitive, physical, and psychological well-being of the teacher [70,71]. Notwithstanding, the results of one

study, [72], investigating gender differences found no statistically significant differences between the metacognitive awareness of male and female teacher trainees.

High levels of EI and MC thus help ensure teachers can interact effectively with both their colleagues and students [1]. Accordingly, a society that fails to adequately prepare teachers to adopt such self-care strategies not only risks sacrificing teachers' well-being and increasing burnout, but also inhibits the development of a positive class climate, good class management, and healthy interpersonal relationships [72].

### 3. Study Aims and Hypotheses

In response to the above-described literature, this study aims to verify whether the dysfunctional outcomes deriving from the need to adapt to new teaching methods that make use of technological instruments can be mitigated by EI and MI as protective factors. Thus, the following hypotheses were proposed.

**Hypothesis 1 (H1).** *Remote work teaching risk factors are positively related to the negative outcomes of burnout and stress.*

**Hypothesis 2 (H2).** *Emotional intelligence mediates the relationship between remote work teaching risk factors and negative outcomes of burnout and stress.*

**Hypothesis 3 (H3).** *Metacognition moderates the relationship between remote work teaching risk factors and emotional intelligence.*

The conceptual model of these hypotheses is depicted in Figure 1.

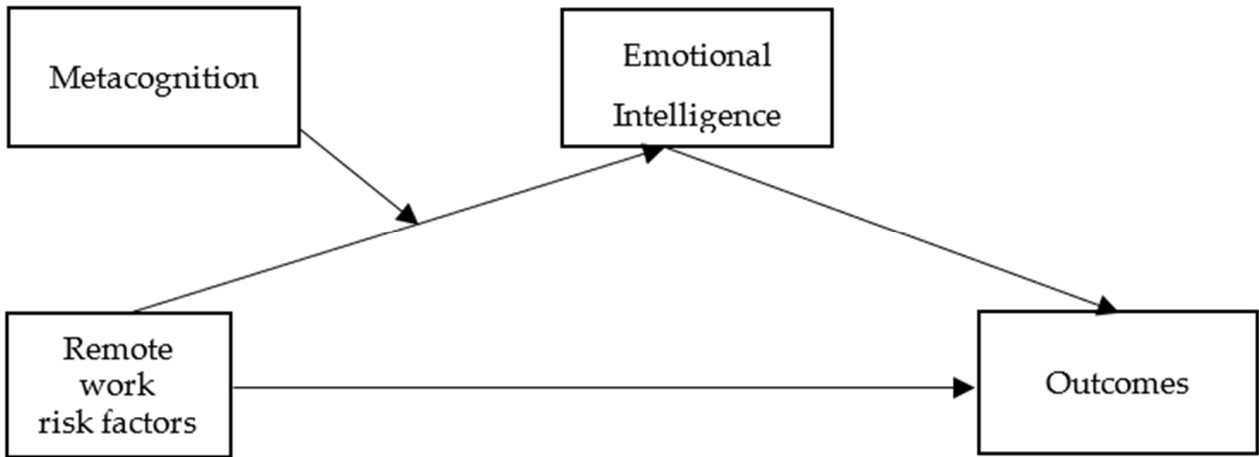

**Figure 1.** Tested theoretical model.

### 4. Materials and Methods

*4.1. Sample and Procedure*

An online questionnaire was administered between April 2020 and December 2020 to 604 Sicilian teachers, who were mostly women ($N = 458$; 75.8%), were mostly married ($N = 377$; 62.4%), had an average age of 47.38 years ($SD = 9.57$), were working in primary ($N = 401$; 66.4%) and elementary schools ($N = 203$; 33.6%), and had been doing so for an average of 16.36 ($SD = 10.48$) years.

All participants gave written informed consent before completing the questionnaire, and all procedures in the present study were performed in accordance with the 1964 Helsinki Declaration, as amended, and/or comparable ethical standards. Furthermore, the Internal Review Board (IRB) of the Faculty of Human and Social Sciences of Kore University of Enna approved the research under protocol number UKE-IRBPSY-04.20.ADD1.

*4.2. Measures*

This study was conducted by incorporating the following scales into the questionnaire:

(1) Remote work teaching risk factors were measured using an ad-hoc scale of six pandemic-time remote work teaching risk factors identified in the literature [33–36], with each factor measured via a 4-point Likert scale from 0 (no risk) to 3 (high risk). These risk factors were (1) a lack of adequate IT tools of exclusive use for online teaching; (2) a lack of computer skills in the use of online educational platforms; (3) a lack of an adequate place of exclusive use for online teaching; (4) a lack of IT skills and time for adapting to online teaching methods; (5) difficulties in managing the class during online teaching; and (6) difficulties in terms of coordination and support among colleagues. The Cronbach's alpha for this scale was calculated as = 0.90.

(2) The Schutte Self-Report Emotional Intelligence Test (SREIT) [73,74] is a self-report instrument designed to measure the emotional skills according to Salovey and Mayer's Emotional Intelligence (EI) model [75]. The SREIT examines 33 self-report items (e.g., "Emotions are one of the things that make my life worth living" and "Some of the major events of my life have led me to re-evaluate what is important and not important") using a 5-point Likert scale (from strongly disagree to strongly agree). The Cronbach's alpha for this scale was calculated as = 0.89.

(3) The Metacognitive Functions Screening Scale (MFSS-30) [76] consists of 30 items measuring four skills: (1) the ability to recognize one's own emotional states and those of others (ARE) (e.g., "I often don't know which adjective to use to describe an emotion of mine" and "There are times when I feel weird, but I don't understand the kind of feeling I'm having"); (2) the ability to recognize causal relationships (ARC) (e.g., "I had conflicts with people close to me, because they did not feel understood by me" and "The others judge me an impulsive person who does not care about the consequences of my own actions on others, though I do not realize"); (3) the ability to decenter, that is, to distance oneself from aspects of a situation (AD) (e.g., "I often find myself unable to 'tune in' with the emotions felt by the people I come into contact with" and "I usually understand very well what a person wants to tell me and his true intentions, regardless of appearances or what he says"); and (4) the ability to ponder situations and problems (AP) (e.g., "When something important happens to me, I usually go over every detail in my mind in order to understand the reasons that behind it" and "When dealing with important or delicate situations, I always try to value previous experiences in order to avoid negative consequences"). The response-scale range has 4 points, from 0 (absolutely false) to 3 (absolutely true). The original coefficient alpha for the scale was 0.79 for ARE, 0.71 for ARC, 0.78 for AD, and 0.70 for AP. The coefficient alpha for the scale used in this study was 0.84 for ARE, 0.86 for ARC, 0.90 for AD, and 0.88 for AP, with a total coefficient alpha value of 0.82.

(4) The Link Burnout Questionnaire (LBQ) [77] measures professional burnout in teachers and others professionals of the helping professions via four sub-scales (with six items each): (1) psychophysical exhaustion (e.g., "I feel physically exhausted from my work" and "During work I feel under pressure"); (2) deterioration of relations with users (e.g., "Compared to other facilities I seem to work with difficult users" and "Users seem ungrateful to me"); (3) job ineffectiveness (e.g., "I feel inadequate in dealing with the problems of my users" and "I feel that my skills are not enough to deal with the unexpected"); and (4) disappointment (e.g., "I think that, if I could start over, I would choose another profession" and "I doubt what I do has any value"). The response scale is a 6-point Likert, from 1 (never) to 6 (every day). The original coefficient alpha was 0.77 for psychophysical exhaustion, 0.69 for deterioration of relations with users, 0.68 for job ineffectiveness, and 0.85 for disappointment. The coefficient alpha for the scale used in this study was 0.90 for psychophysical exhaustion, 0.90 for deterioration of relations with users, 0.90 for job ineffectiveness, and 0.91 for disappointment, with a total coefficient alpha value of 0.96.

(5)　The Secondary Traumatic Stress Scale STSS-I [78,79] investigates indirect reactions to traumatic experiences using a 15-item Likert scale, from 1 (never) to 5 (every often). The original version consists of 17 items. In the Italian version, items 12 and 13 have been removed because they were not normally distributed. STSS-I has two sub-scales: (1) arousal (AR) (e.g., "I felt emotionally numb" and "I felt easily irritable"); and (2) intrusion (IN) (e.g., "My heart beat faster when I thought of my work with the victims" and "I felt like I was reliving the traumas experienced by the victims"). The original coefficient alpha was 0.87 for AR and 0.81 for IN. In this study, the coefficient alpha was 0.91 for AR and 0.63 for IN, with a total coefficient alpha value of 0.59.

(6)　An ad hoc Socio-Demographic Questionnaire was used, via which participants were asked to provide additional information on socio-demographic characteristics, such as age, gender, marital status, seniority, and teaching subject.

### 4.3. Data Analysis

The difference between groups in terms of the outcome levels for each of the remote work risk factors were tested via *t*-tests, while the effect of socio-demographic variables was analyzed via independent sample *t*-tests, ANOVA, and correlational analysis, using IBM SPSS 22.0.

Two moderated mediation model analyses were conducted to test whether the effect of remote work risk factors (as independent variables) on outcomes (the dependent variables of burnout and secondary traumatic stress) was mediated by emotional intelligence with meta-cognition acting as a moderator between the risk factors and emotional intelligence (PROCESS Model Number 7, with 5000 re-samplings) [80].

Using a single questionnaire for all variables with numerous single-item measures, the common method variance was considerably limited, as per the literature on common method biases in behavioral research [81]. In addition, the different scales were randomly inserted into the questionnaire and were graphically separated from each other. Finally, different scale endpoints and formats for the measures were used in order to further mitigate method biases attributable to commonalities in scale endpoints and anchoring effects.

### 5. Results

Preliminary analyses were performed to ensure there was no violation of the assumption of normality, linearity, and multicollinearity. Multivariate outliers were identified using the Mahalanobis distance, and the sample was reduced to 604 subjects.

No correlations between age or seniority and any of the other sociodemographic variables with any of the measured variables were found. Furthermore, no gender differences were found for any of the measured variables.

Independent sample *t*-tests revealed significant differences in the negative outcomes (burnout and stress) between individuals belonging to the group with an absence of the risk (risk level = 0) or presence of the risk (risk level = 2–3) regarding each remote teaching risk factor (see Table 1).

In order to test the research hypotheses (i.e., whether the effect of remote work risk factors on outcomes is mediated by emotional intelligence, with metacognition acting as a moderator), two moderated mediation model analyses (PROCESS Model Number 7, Figure 1) were calculated.

With burnout as the dependent variable, the overall equation was significant, $R2 = 0.71$, $F (3, 600) = 713.9$, $p < 0.001$, while the direct effect of the remote work risk factors on burnout was slightly significant, and the indirect effect on burnout through emotional intelligence was significant and moderated by metacognition ($R2$ change $= 0.01$, $F (1, 600) = 21.58$, $p < 0.000$) (Figure 2). The mediation effect of the relationship between risk factors and burnout was significant for low levels of metacognition ($t = -5.28$, $p < 0.000$; $LL = -0.459$; $UL = -2.10$), significant for medium levels ($t = -2.34$; $p < 0.019$; $LL = -1.42$; $UL = -0.12$), and not significant for high levels of metacognition ($t = 1.24$; *n.s.*; $LL = -0.32$; $UL = 1.43$).

**Table 1.** Difference in negative outcomes between groups with the absence or presence of remote work teaching risks.

| | Burnout | | Stress | |
|---|---|---|---|---|
| **Risk Factor** | **Absent** | **Present** | **Absent** | **Present** |
| Lack of a dedicated and exclusive PC | 12.7 (4.74) | 20.8 *** (9.01) | 2.4 (0.58) | 2.9 *** (0.61) |
| Unfamiliarity with IT platforms | 12.4 (4.1) | 21.9 *** (8.9) | 2.4 (0.58) | 2.9 *** (0.53) |
| Lack of an exclusive place for remote teaching | 12.9 (4.95) | 19.6 *** (9.1) | 2.4 (0.59) | 2.7 *** (0.67) |
| Need to adapt to online teaching methods | 15.4 (7.58) | 12.9 *** (4.7) | 2.6 (0.63) | 2.4 *** (0.61) |
| Difficulties in class management | 13.4 (5.94) | 17.5 *** (8.45) | 2.5 (0.59) | 2.6 * (0.68) |
| Difficulties in coordination with colleagues | 18.9 (8.89) | 12.5 *** (4.38) | 2.8 (0.66) | 2.4 *** (0.57) |

\* $p < 0.05$; *** $p < 0.001$.

The correlations among the measured variables and descriptive statistics are presented in Table 2.

**Table 2.** Descriptive statistics and correlations among the measured variables.

| | **Mean (SD)** | **1** | **2** | **3** | **4** | **5** |
|---|---|---|---|---|---|---|
| 1. Risk factors | 1.48 (0.75) | - | | | | |
| 2. Burnout | 14.89 (7.1) | 0.339 *** | - | | | |
| 3. Secondary traumatic stress | 2.53 (0.63) | 0.310 ** | 0.644 *** | - | | |
| 4. Emotional intelligence | 121.34 (22.15) | −0.395 *** | −0.838 *** | −0.479 *** | - | |
| 5. Metacognition | 59.25 (17.68) | −0.288 *** | −0.781 *** | −0.486 *** | 0.842 *** | - |

** $p < 0.01$; *** $p < 0.001$.

With stress as the dependent variable, the overall equation was significant, $R2 = 0.23$, $F (3, 600) = 89.42$, $p < 0.001$, while the direct effect of remote work risk factors on stress was slightly significant, and the indirect effect on stress through emotional intelligence was significantly moderated by metacognition ($R2$ change = 0.01; $F (1, 600) = 21.58$; $p < 0.000$) (Figure 3). The mediation effect of the relationship between the risk factors and stress was significant for low levels of metacognition ($t = −5.28$; $p < 0.000$; $LL = −0.459$; $UL = −2.10$), significant for medium levels ($t = −2.34$; $p < 0.019$; $LL = −1.42$; $UL = −0.12$), and not significant for high levels of metacognition ($t = 1.24$; *n.s.*; $LL = −0.32$; $UL = 1.43$). Figures 4 and 5 show the conditional effect of the remote work risk factors on the different outcomes mediated by metacognition and moderated by emotional intelligence.

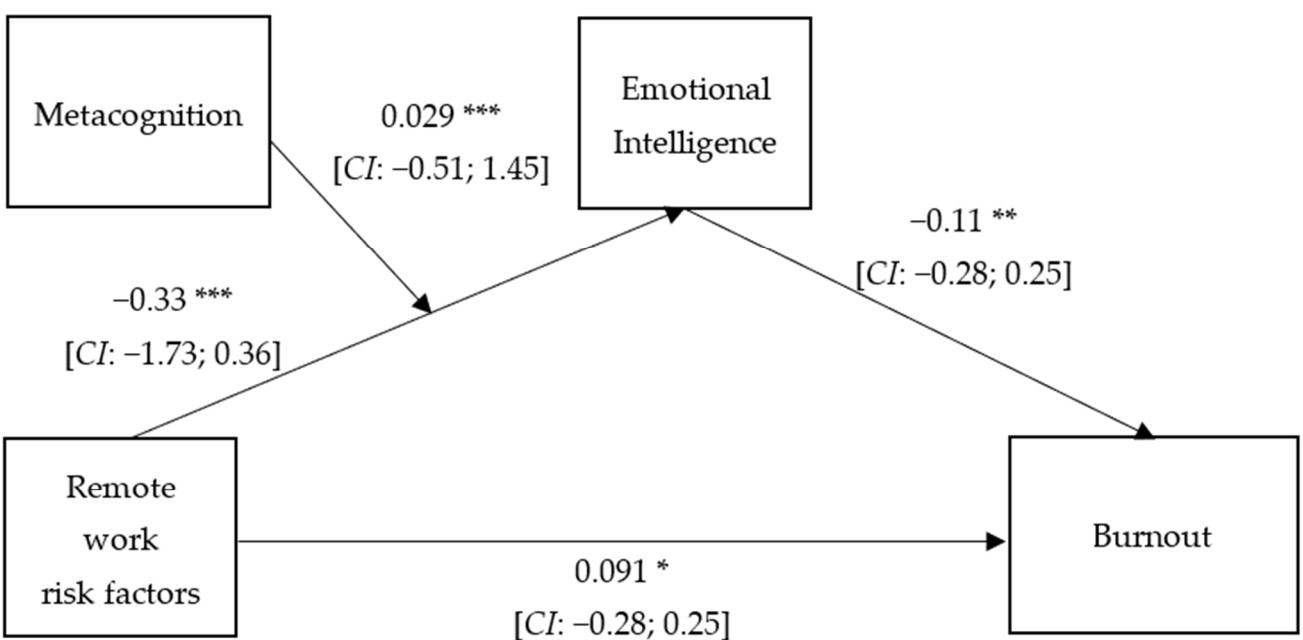

* $p < 0.05$; ** $p < 0.01$; *** $p < 0.001$

**Figure 2.** Moderated mediation analysis with burnout as the outcome.

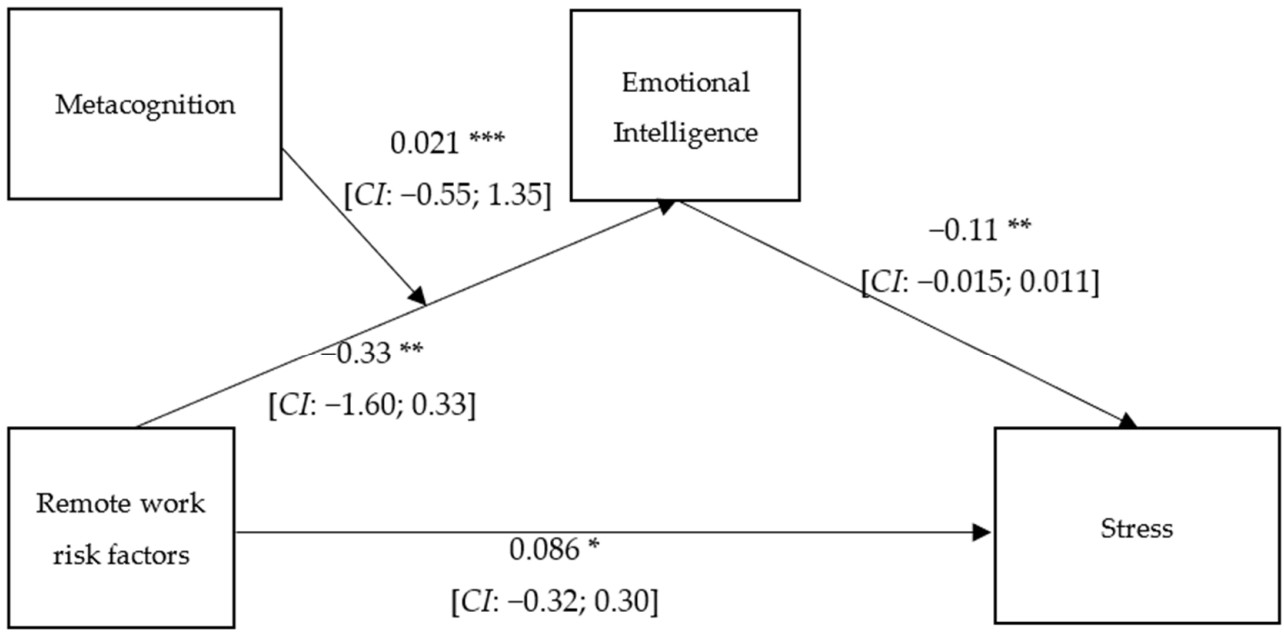

* $p < 0.05$; ** $p < 0.01$; *** $p < 0.001$

**Figure 3.** Moderated mediation analysis with burnout as outcome.

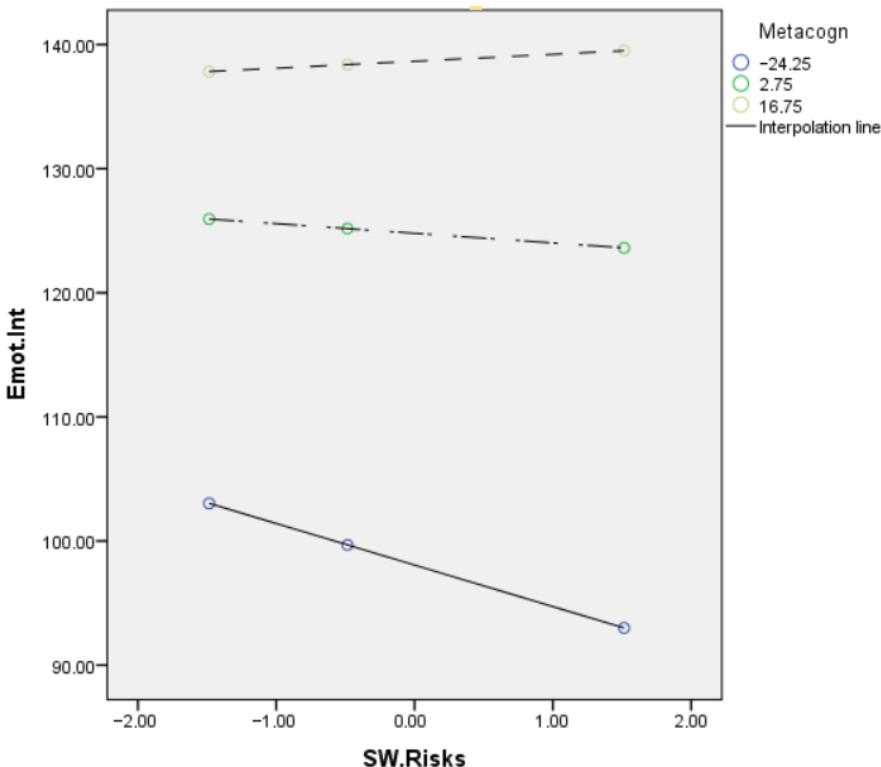

**Figure 4.** Conditional effect of the remote work risk factors on burnout mediated by metacognition and moderated by emotional intelligence.

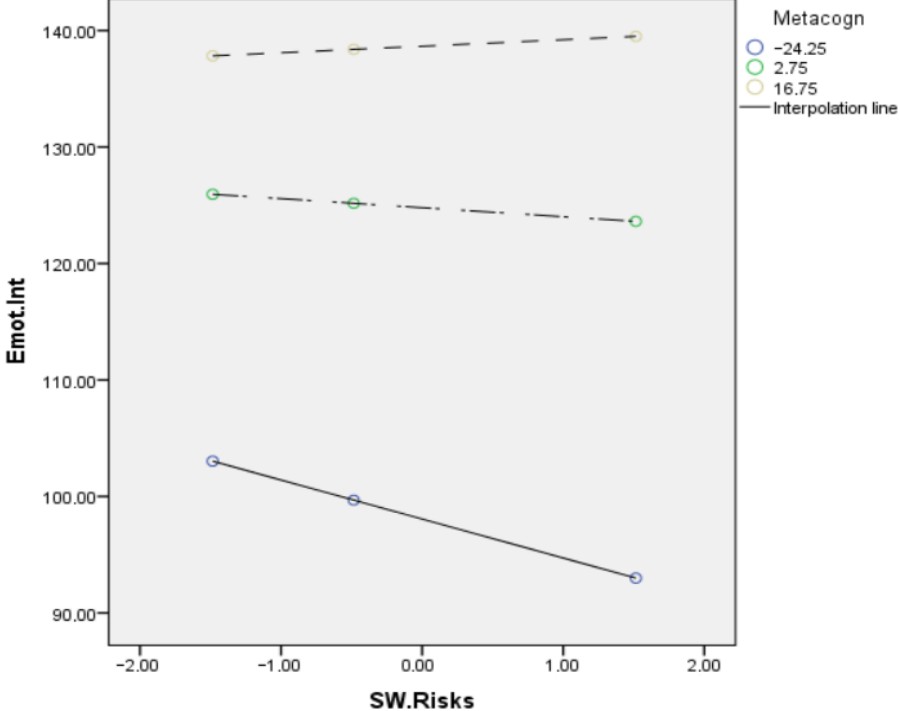

**Figure 5.** Conditional effect of the remote work risk factors on stress mediated by metacognition and moderated by emotional intelligence.

## 6. Discussion

The aim of the present study was to analyze whether the dysfunctional outcomes of burnout and stress due to the need to adapt to new teaching methods via remote work

technological instruments can be mitigated by emotional intelligence and metacognition as protective factors.

The literature [21,28–30] suggests that certain risk factors associated with remote teaching (lack of a dedicated and exclusive PC, unfamiliarity with IT platforms, lack of an exclusive place for remote teaching, the need to adapt to new online teaching methods, difficulties in class management, difficulties in coordination with colleagues) may correlate with emotional intelligence, metacognition, burnout, and secondary traumatic stress.

In this study, correlation analyses confirmed a statistically significant positive correlation ($p < 0.001$) between the considered risk factors and burnout, along with a further statically significant positive correlation ($p < 0.01$) with secondary traumatic stress. This indicates that, in the presence of the risk factors considered in this study, teachers are more likely to experience burnout and stress. Conversely, the risk factors showed a statistically significant negative correlation ($p < 0.001$) with emotional intelligence and metacognition; although the presence of the risk factors decreased the effect of the protective factors of emotional intelligence and metacognition, as suggested by the literature [17,43,66,72].

Based on these results, two moderated mediation analysis models were hypothesized: one model with burnout as an outcome and one model with stress as outcome. In the first model, the direct relationship between the effects of risk factors on burnout was found to be mediated by emotional intelligence. Moreover, metacognition was found to moderate the relationship between the effect of risk factors on emotional intelligence. This suggests that if a teacher possesses high levels of emotional intelligence and has hypothesized the risk factors, their emotional intelligence decreases the likelihood of burnout [29,43,54,55]. In our model, the mediation effect of the relationship between risk factors and burnout was statistically significant for low levels of metacognition, significant for medium levels, and not significant for high levels. This relationship was tested again in the second model, but with the outcome of secondary stress. The mediation effect of the relationship between risk factors and stress was thus found to be statistically significant for low levels of metacognition, significant for medium levels, and not significant for high levels. Our models are, therefore, in agreement with studies investigating the relationship between burnout and emotional intelligence [54,55]. Indeed, EI is believed to play a crucial role in teachers' occupational health models [57], and its mediating role has been confirmed by other studies [58,59] that suggest that the protective factors of emotional intelligence and metacognition help prevent the emergence of clinical symptoms.

The literature [54,63] regarding the teaching profession is in accordance with our moderation–mediation models, which confirm that high EI levels mediate the relationship between risk factors, burnout, and stress [64,82]. Several studies [31,68,69] that explore the relationship of metacognition with stress report its moderating function, while others suggest that metacognitive strategies can predict traumatic stress symptoms and burnout syndrome [52,53], and these two aspects are confirmed by our study.

With regard to Target 4.7 of the United Nation's Agenda 2030 Sustainable Development 4.7 Goals, these results highlight the need to re-think educational models for adequate teacher training, with repercussions for the quality of education, psychological well-being, and the future development of teachers' professional careers [32,39,57,70,83].

*Limitations and Future Directions*

The current study has certain limitations that should be considered in future research. First, the study is limited in terms of sample characteristics. A larger sample would have provided a more complete picture of the causes of stress, the role of teachers' stress protective factors in remote working, and possible interventions. Even if the sample is completely in line with the average age and seniority of the population, problems of external validity should be considered in light of the differences between Italian teachers and those in other geographical contexts, where the average age and seniority are generally lower. There is little doubt that these factors could play a role in some of the measured variables. Second, future research should include a more detailed analysis of contextual factors, such

as the school leader's leadership role and the collaborative climate among colleagues [84,85], in order to better understand how teachers' evaluations of internal support may depend on the quality of the work environment. Third, longitudinal studies are needed to determine whether internal and external support available to teachers determines and/or influences their emotional intelligence. The results of this study should, therefore, be considered to be a preliminary assessment for a larger intervention study to evaluate potential prevention programs that can be implemented for stressed teachers [21,29,40,67,86].

## 7. Conclusions

Theoretical perspectives on EI suggest pathways to enhance teachers' personal resources and thus their ability to cope with stressful events [87]. Stress in the educational environment not only affects teachers, but also has negative effects on the educational institution, and now, more than ever, stressors are believed to be important in predicting burnout outcomes [12,13].

Our results confirm the idea that emotional intelligence and metacognition play a significant role in helping teachers manage the stress associated with remote working and thus can mitigate burnout and other dysfunctional symptoms that result from difficulties in adapting to new ways of teaching through remote work technological tools [88]. The findings are particularly relevant for lockdown scenarios, where maintaining contact with young students through remote teaching can be an emotionally stressful context that can compromise well-being outcomes and cause negative impacts on teachers' well-being and job performance [13].

Many empirical and theoretical contributions [30,37] assert that healthy behavior outcomes have an important effect on work performance, productivity, and relationships, and such contributions help us better understand the practical relevance of teachers' health statuses, which in turn can also affect students and learning outcomes.

Our results led us to consider implications for the professional development of teachers, such as the need to identify dimensions of stress related to remote teaching, and to further explore variables, including protective factors for managing stress in remote work teaching situations or organizational factors, such as leadership or climate [89], that should be included in future studies.

**Author Contributions:** Conceptualization, C.I., B.C., R.I., E.M.C.L. and G.F.; methodology, T.R. and M.B.; formal analysis, M.B.; investigation, B.C., E.M.C.L. and G.F.; data curation, G.F., E.M.C.L. and M.B.; writing—original draft preparation, T.R., B.C. and M.B.; writing—review and editing M.B., B.C., R.I. and T.R.; writing–results section, M.B. and T.R., writing–discussion section, B.C. and R.I.; supervision, T.R., C.I. and M.B. All authors have read and agreed to the published version of the manuscript.

**Funding:** This research received no external funding.

**Informed Consent Statement:** Informed consent was obtained from all subjects involved in the study.

**Data Availability Statement:** Not applicable.

**Conflicts of Interest:** The authors declare no conflict of interest.

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
