# Peer review of "The Role of Emotional Intelligence and Metacognition in Teachers’ Stress during Pandemic Remote Working: A Moderated Mediation Model"

_ejihpe, doi:10.3390/ejihpe13010006_

Round 1
Reviewer 1 Report
This study is well-written and competently executed. It provides some new information regarding teachers´ experiences of the Covid pandemic.
The issue with these types of studies is the external validity. It is thus hard to generalize based on the results presented in the paper. It is not very much the authors can do about that, apart from providing more information about how the sample used in the analysis compared to teachers in Italy in general. For instance, teachers in the sample seem to be quite old, and have a long job tenure. Is this very different to teachers in Italy in general. Would the results be different if more young teachers would be included?
Author Response
- Comment: This study is well-written and competently executed. It provides some new information regarding teachers´ experiences of the Covid pandemic.
Response: Really thanks for the appreciation. We are really happy with your opinion, and we are honored.
- Comment: The issue with these types of studies is the external validity. It is thus hard to generalize based on the results presented in the paper. It is not very much the authors can do about that, apart from providing more information about how the sample used in the analysis compared to teachers in Italy in general. For instance, teachers in the sample seem to be quite old, and have a long job tenure. Is this very different to teachers in Italy in general. Would the results be different if more young teachers would be included?
Response: We believe that your comment is appropriate and that we must take it into account for the improvement of the paper. For information, however, we would like to point out that our sample is totally in line, both in terms of average age and seniority, with the population of primary and secondary school teachers in Italy. You will be able to find out from the OECD data that Italian school teachers are those in Europe with the highest average age and seniority. However, we believe your advice is correct, because this information is not clear to the international reader. Therefore we have added the following sentence in the limitations section:
“Even if the sample is completely in line with the average age and seniority of the population, problems of external validity should be considered in light of the differences between Italian teachers and those in other geographical contexts, where the average age and seniority are generally lower. There is little doubt that these variables could play a role in many of the variables measured”

Reviewer 2 Report
The article examined the influential effects of EI and MC on teacher stress due to a switch to online learning. The article is well-written and the results are unsurprising. The methodology is straightforward. I have a couple of comments:
1. There seems to be an error on lines 362 and 366 - I think the authors meant to say "high" levels of metacognition first, because they say "low" later, e.g., I think they're trying to differentiate between high, medium and low levels but say "low" twice.
2. That gets to a concern of mine - I don't think it's correct to say "highly" significant vs significant - data differences are significant or not. I believe that the authors are implying that level of significance equates to "effect size" and that's simply not true - the manuscript should be altered as to not imply that.
Author Response
Comment: The article examined the influential effects of EI and MC on teacher stress due to a switch to online learning. The article is well-written and the results are unsurprising. The methodology is straightforward.
Response: Thank you so much for the overall rating.
I have a couple of comments:
- Comment: 1. There seems to be an error on lines 362 and 366 - I think the authors meant to say "high" levels of metacognition first, because they say "low" later, e.g., I think they're trying to differentiate between high, medium and low levels but say "low" twice.
Response: Thank you for the suggestion. We corrected as suggested.
- Comment: 2. That gets to a concern of mine - I don't think it's correct to say "highly" significant vs significant - data differences are significant or not. I believe that the authors are implying that level of significance equates to "effect size" and that's simply not true - the manuscript should be altered as to not imply that.
Response: Thank you for the suggestion. We corrected it as suggested. We deleted “highly”.
Thank you really much for your help.
